# Tactile Object Recognition for Humanoid Robots Using New Designed Piezoresistive Tactile Sensor and DCNN

**DOI:** 10.3390/s21186024

**Published:** 2021-09-08

**Authors:** Somchai Pohtongkam, Jakkree Srinonchat

**Affiliations:** Department of Electronics and Telecommunication Engineering, Rajamangala University of Technology Thanyaburi, Khlong Luang 12110, Thailand; somchai_po@mail.rmutt.ac.th

**Keywords:** tactile sensor, tactile object recognition, DCNN, humanoid robot, transfer learning

## Abstract

A tactile sensor array is a crucial component for applying physical sensors to a humanoid robot. This work focused on developing a palm-size tactile sensor array (56.0 mm × 56.0 mm) to apply object recognition for the humanoid robot hand. This sensor was based on a PCB technology operating with the piezoresistive principle. A conductive polymer composites sheet was used as a sensing element and the matrix array of this sensor was 16 × 16 pixels. The sensitivity of this sensor was evaluated and the sensor was installed on the robot hand. The tactile images, with resolution enhancement using bicubic interpolation obtained from 20 classes, were used to train and test 19 different DCNNs. InceptionResNetV2 provided superior performance with 91.82% accuracy. However, using the multimodal learning method that included InceptionResNetV2 and XceptionNet, the highest recognition rate of 92.73% was achieved. Moreover, this recognition rate improved when the object exploration was applied to demonstrate.

## 1. Introduction

Unlike humans who can identify objects by touching, humanoid robots do not have this capability due to the lack of suitable tactile sensors and efficient recognition processing systems. The critical development of humanoid robot technology can be divided into two parts: (1) robot anatomy [1]; (2) the robot nervous system [2]. Developing a physical structure and human-like learning ability is necessary to enable robots to operate in a home or office environment. In addition, the development of robots having a human-like hand structure is desirable [3,4,5]. This study examines a humanoid robot’s physical sensory system that can recognize objects by touch. Its essential function is developed based on the human physical sensory system [6]. In object learning and recognition systems of humanoid robots employed artificial haptic perception [7,8,9,10,11], pressure sensors or tactile sensors are utilized [7,8,9,10,11], and the obtained information is sent to a computer for analysis [10]. Object learning and recognition systems are similar to the human sensory system where nerve-ending receptors (e.g., Ruffini endings and Pacinian receptors) obtain information sent to the brain for interpretation. There have been numerous studies describing the development of robotic hands. These studies focus on tactile sensor arrays for robot hand artificial skin application [7,8,9,10,11]. Human sensory recognition is a complicated action resulting from the biosensor system in the body, which includes three modes of data perception [6]. The first mode is tactile perception where contact with the skin of the fingers or palm provides information on the contact geometry or pressure profile.

A tactile sensor array produces this mode of data perception for robots and presents data in a 2D format or tactile image [10]. The second perception mode is kinesthetic perception, a perception from motion such as rubbing or scrubbing objects. For robots, this mode of data perception is produced by tactile sensors on the fingertips or palm from dragging the sensor onto the object and presents data in a 1D format [12]. The third perception mode is global object shape, where perception data is gathered through the joints, and the obtained data are used to indicate the global object shape or geometry. This mode of data perception is produced by a bend sensor or angle sensor on the finger joints [13]. The development of artificial haptic perception for integration with a robot hand is challenging because it must mimic the touching of human hands resulting from dexterous movement. The touching of an object provides information on object properties that other methods cannot estimate; for example, we cannot evaluate an object’s softness, roughness, and smoothness without touching it. Another advantage of touch is that it provides data if the visual sensory system fails. Therefore, the haptic sensory system is important in the context of multisensory perception. Humans can distinguish objects using only haptic perception. Still, current robots lack this skill, mainly due to the lack of suitable tactile sensors and appropriate methods for interpreting the resulting data. In this study, a sensor array of a proper size for fixing to the humanoid robot hand and the algorithm for effective tactile object recognition were developed to efficiently operate the humanoid robot object recognition system.

## 2. Related Work

This research involved two main parts: (1) the design of the tactile sensor array and (2) the development of the algorithm for object recognition based on tactile image recognition for humanoid robots.

### 2.1. Tactile Sensor Array

A tactile sensor array can be developed with several operating principles, and it has various shapes and sizes depending on applications [7,8,9,10,11,14,15,16,17,18,19,20]. There are many types of tactile sensors, which can be categorized by the working principles such as piezoresistive [21,22], capacitive [23,24], piezoelectric [25,26], and optical [27,28], etc. This research focused on designing the tactile sensor array based on the piezoresistive principle due to the simple structure, high sensitivity, low cost, and robustness [9,10,16,20,21]. This type of tactile sensor array is commonly used in various applications such as medical [5,7,20], industrial manufacturing [29], civil engineering [30], and human-like activities [31], etc. Recently, the application of piezoresistive tactile sensors for humanoid robots has been gaining much interest [1,2,5,8] [20,21,22,23,24,25,26,27,28,29,30,31]. The working principle of the piezoresistive tactile sensor can be explained by the piezoresistive effect [16,21,22]. The electrical resistance is affected when an object’s boundary surface changes, as described in Equation (1) [16].
(1)$$\frac{\Delta R}{R}=\left(1+2\sigma +\pi E\right)\chi$$
(2)$$R=p\frac{L}{A}$$
where *R* is the resistance of conducting material along the length *L*, Δ*R* is the change of resistance induced by strain on the conductor, *σ* is the Poisson’s ratio of the material, *π* is the piezoresistive coefficient, *E* is Young’s modulus, and *χ* is the strain caused by an acting force. The conductor resistance can be calculated by Equation (2). Table 1 shows a variety of tactile sensor arrays that have been developed for a humanoid robot. A distinctive tactile sensor array design with a simple PCB structure was applied for the humanoid robot finger [32]. However, its asymmetrical pixel design and geometry were not suitable for the humanoid robot palm. A high-resolution tactile sensor was designed for Flex e-Skin, but the required size was a trade-off [33]. Many tactile array sensors were developed for touch sensing and Flex e-Skin [32,33,34,35,36,37,38,39,40,41,42,43,44], while the resolution was compromised. Hence, they were not suitable for developing tactile sensor arrays for high accuracy image recognition capability.

A simple design structure of a conductive rubber transducer was developed using the first-order feature and k-nearest neighbor (kNN) to improve the object recognition capability via the sense of touch [36]. In this case, a gripper was used as fingers; hence, the sensor was too small for the humanoid robot palm [32,36,39,43,44]. On the other hand, many piezoresistive tactile sensors have also been developed with a relatively large physical size unsuitable for applying to the humanoid robot palm [33,37,41,42]. Although a tactile sensor with a proper size was proposed [35], there is an issue with developing the algorithm for object recognition. Therefore, our research focuses on the new design of tactile sensors in terms of resolution and the physical size that can be efficiently applied to the humanoid robot palm. The algorithm for the tactile image recognition method was also developed.

### 2.2. Tactile Image Recognition for the Humanoid Robot

The humanoid robot hand has been developed with different types of sensors for various applications such as medical [5,45,46], industrial [8,30], and home use applications [5]. One of the significant developments in data processing is the tactile sensor for the object recognition capability of robots [36,39,47,48,49,50,51,52,53,54,55,56,57,58,59]. Object recognition for humanoid robots has been reported in various research studies [10,16,18], relying on the humanoid robot hand. The essential element for processing is the installation data of the tactile sensor array, so-called tactile image recognition, as summarized in Table 2.

Earlier developments of tactile image recognition were based on the first-order feature [36,47,49,53], and the object orientation was not considered [36,47]. The bag-of-words (BoW) technique with high order descriptor was applied to obtain high accuracy in object recognition [48,52]. However, each object has to be touched by the sensor several times for the recognition system to learn about that object gradually. Alternatively, a combination of the Haar Wavelet method and the kNN method was employed to develop a recognition system to learn about each object by a single touch [51]. However, this was not applied in the form of a humanoid robot palm. A kernel principal component analysis (K-PCA) method and multiple kernel learning (MKL) algorithm with a support-vector machine (SVM) method were applied for the low-resolution tactile sensor. These methods can improve the object recognition rate for the low-resolution tactile sensor for a single touch [39]. However, the objects used in the experiment were small-sized, and the object orientation was not considered. Cretu et al. [55] used sensors to improve the object recognition rate; however, this method is not compatible with the image processing technique. A technique called kernel sparse coding (KSC) was proposed to shorten the calculation time. At the same time, the orientation independence in the learning procedure was achieved by using three sets of tactile sensors [55]. A method called iterative closest labeled point (iCLAP) was employed to develop a recognition system to learn about objects [56]. However, each object has to be touched by the sensor several times for the recognition system to learn about that object gradually. Albini et al. [57] applied a high-resolution sensor (768 pixels) with an AlexNet-deep convolution neural network (AlexNet-DCNN) to achieve a high recognition rate compared to that of BoW. The recognition rate was further improved upon increasing the resolution of the tactile sensor (1400 pixels) using AlexNet-DCNN [58]. The same hardware was tested with ResNet-DCNN, and an improved recognition rate was obtained [59].

According to previous research, BoW can be used with low-resolution sensors (36 pixels [48], 84 pixels [47,52]), and the physical size was suitable for installation at the fingertip. Although DCNN can work well with high-resolution sensors (768 pixels [57], 1400 pixels [58,59]), the large physical size was impractical for installation on the humanoid robot palm. In the present research, the tactile sensor array (256 pixels) has been developed with a physical size suitable for the application of the humanoid robot palm (56.0 × 56.0 mm). Various DCNNs (19 models) with transfer learning methods were applied for tactile image object recognition. The appropriate DCNN was chosen for further improvement via resolution enhancement. Multimodal learning methods and object exploration were also implemented to increase the recognition rate effectiveness.

## 3. Methodology and Experimental Setup

Figure 1 represents an overview of the contribution of this work. The block diagram proposed a tactile object recognition process using DCNN. This work presents the development of a tactile sensor array with high sensitivity and a suitable size for the humanoid robot hand, which it then mounts on the humanoid robot hand and randomly positioned object capture is carried out. Then, tactile image data is obtained by capturing objects of each class as a data set for testing with a recognition system. Tactile images are tested with recognition system and resolution enhanced by bicubic interpolation method to increase efficiency. The recognition algorithm is tested using transfer learning DCNN comparing 19 models, optimized using multimodal approach and multiple times of handling by object exploration method. The details of each part of the work are as follows.

### 3.1. Sensor Design and Result

Most electrode designs with small sensors are circle electrodes, as shown in Figure 2 [35], with many dead areas (black area). It leads to a loss of information. This research designed a square electrode instead of the circle electrodes to reduce the dead area. The contact area of the circle electrode can be calculated using Equation (4) where *R* is the radius. The contact area of the square electrode can be calculated with Equation (5) where *d* is the width.
(3)$$A=A_{1\;}-A_{2\;}+A_{3\;}-A_{4}$$

Therefore, the active area of the circle electrode is
(4)$$A=\pi R_{1}^{2}-\pi R_{2}^{2}+\pi R_{3}^{2}-\pi R_{4}^{2}$$

The active area of the square electrode is
(5)$$A=d_{1}^{2}-d_{2}^{2}+d_{3}^{2}-\pi R_{4}^{2}$$

The contact area of the square electrode sensor is 8.3692 mm^2^, which is 21.45% larger than that of the circle electrode sensor (6.5665 mm^2^). Moreover, it provides higher sensitivity to the sensor.

The active area of the circle electrode sensor when the sensor is partially touched is shown in Figure 3 where *R*_1_ and *R*_3_ are the radius of the outer and inner electrodes, respectively. *R*_2_ and *R*_4_ are the radius of the gap and the through-hole, respectively. Moreover, *h*_1_ and *h*_3_ are the contact distance of the outer and inner electrodes, respectively. *h*_2_ and *h*_4_ are the contact distance of the gap and the through-hole, respectively. These areas can be calculated by using Equations (6) and (7), and the total area can be calculated by using Equation (3).
(6)$$A=\frac{R^{2}}{2}\left(\frac{\alpha \pi }{180}-sin\frac{\alpha \pi }{180}\right)$$
(7)$$\alpha ={2cos}^{-1}\frac{\left(R-h\right)}{R}$$
where *R* is the radius and *h* is the contact distance.

Figure 4 shows the active area of both circle and square electrode sensors when the contact comes from different directions. Each sensor’s active area per %contact in the circle and the square electrode can be calculated with Equations (3), (5), and (6), respectively. The square electrode sensor offers a higher active area per %contact when the contact exceeds 10%. This difference also increases with increasing %contact, as shown in Figure 5.

The piezoresistive sensor used in this research was conductive polymer (capLINQ, MVCF-40012BT50KS/2A). The piezoresistive element was 0.1 mm thick with a surface resistance of 50,000 ohm/cm^2^. The PCB technology was applied to fabricate sensor structure using epoxy PCB as substrate with a thickness of 0.5 mm. The electrode made of Cu with a thickness of 0.2 mm was gold-plated with a thickness of 18 μm. The electrodes were designed in the form of 16 rows × 16 columns matrix. The square shape can help reduce the dead area as represented in Figure 6a with the 3.5 × 3.5 mm pixels. The external and internal size of the electrode was 3.0 × 3.0 mm and 1.4 × 1.4 mm, respectively. The gap between the internal and external electrodes was 0.1 mm, and the distance between each pixel was 0.5 mm. The size of the sensor array was 56.0 × 56.0 mm as shown in Figure 6a. Figure 6e illustrates the sensor layers where the lowest layer is the substrate functioning as the base and insulator. The next layer is the electrode layer, functioning as a conductor. The third layer is the conductive polymer that functions as Piezoresistive with variable resistance depending on the acting forces. The top layer is the elastic overlay that functions as a receptor and a force transmitter (the conductive polymer).

The sensor developed was tested for two crucial properties: the resistance induced by an acting force and tactile image acquisition upon contacting an object. These were implemented in different sets of experiments, as detailed in the following sections.

#### 3.1.1. Sensor Resistance and Sensitivity

Figure 7 shows the equivalent resistance of a tactile sensor as given by Equation (8).
(8)$$R_{eq}=R_{inn}+R_{vol}+R_{outt}$$
where *R_inn_* and *R_outt_* are the resistances at the interfaces with the inner and outer electrodes, respectively, and *R_vol_* is the path’s resistance between both electrodes. The resistance depends on the effective contact area and pressure. It is proportional to the electrode area. Therefore, we can write
(9)$$R_{inn}=\frac{a\left(P\right)}{A_{inn}}\;;\;R_{outt}=\frac{a\left(P\right)}{A_{outt}}$$
where *A_inn_* and *A_outt_* are the areas of the inner and outer electrodes, respectively, and *a*(*P*) is a function of the pressure exerted on the sensor.

The sensitivity of our sensor was evaluated by the resistance measured at different strengths of an acting force using the apparatus, as shown in Figure 8b. The equipment included a force gauge, a transmission rod with a known cross-sectional area, and an ohmmeter for measuring the resistance. Since the sensor consisted of 256 individual elements, as shown in Figure 8a, the testing positions in the study were categorized into three groups as (1) elements in the corners (C1, C2, C3, and C4. 2), (2) elements on the side (S1, S2, S3, and S4), and (3) elements in the middle of the array (M1, M2, M3, and M4). The test results are presented in Figure 9a–c for different groups of elements. It was observed that the resistance decreased upon increasing pressure, and the difference in resistance among all groups was slight. Figure 9d shows the average resistance of the sensors from all three groups.

Normal distribution was assumed for the resistance measurements for each sensor. The measurements were statistically analyzed and modeled. The mean ($\bar{x}$) and standard deviation (*SD*) were calculated. These were used for quality control in developing sensors. The figure presents the sensors’ measurements in 12 points: C1, C2, C3, C4, S1, S2, S3, S4, M1, M2, M3, and M4 using pressure from 0 kPa to 300 kPa.

Figure 9a–c illustrates distributions of statistical data of sensors measured from the corners, the side, and the middle of sensors with 12 points: C1, C2, C3, C4, S1, S2, S3, S4, M1, M2, M3, and M4. Figure 9d: the blue lines are the mean value of the measurements, and the red lines are the bands that cover 99% of the measurement or ±3*SD*. Moreover, the power formular of nonlinear regression is exploited to calculate the pressure and sensor resistance relationship. The equation is *R = a**P^b^*, where *P* and *R* are the pressure and sensor resistance, respectively. All resistance values were calculated from Equation (10).
(10)$$R=228\times P^{-0.97}$$
where *R* is resistance in Ω, and *P* is the acting pressure in kPa.

#### 3.1.2. Sensor Hysteresis and Reproducibility

Sensor hysteresis was tested with pressure from 0 kPa to 300 kPa by increasing 10 kPa each step, and then reversed back to 0 kPa by decreasing 10 kPa each step. C1 was the position that measured the hysteresis. This found that the sensor had obtained, slightly, potential energy from the previous pressure, as shown in Figure 10a. Sensor reproducibility was tested with pressure 100 kPa. The resistance sensor was measured by loading and unloading the pressure every minute. This found that the resistance sensor slightly decreased, as shown in Figure 10b. Therefore, the amount of hysteresis and reproducibility in the sensor does not directly affect the pattern of object profile, which is the aim of applying the tactile sensor.

### 3.2. Humanoid Robot Hand with Sensor

In this study, a mechatronic robot hand with a human hand-like structure was developed. It consisted of five fingers and a palm, the basic design of a modern humanoid robot. A tactile sensor array was employed as a physical sensory system (see Figure 11a). The sensory system consisted of 3 layers: support, sensor, and elastic layer (3-mm-thick foam sheet). An essential part of the tactile sensor array is the upper surface area used to distribute the pressure force, referred to as the skin layer (see Figure 11c). When an object touches the skin layer, the pressure force is transmitted and distributed onto the tactile sensor array by the elastic material. This slightly transforms depending on the object’s surface. This performance provides a pressure profile result and the object geometry.

### 3.3. Haptic Random Exploration and Objects

The collection of sample images was performed by randomized sampling at different positions of the object. The images obtained from touching were collected from different object positions. There were two main items involving the image acquisition; randomized object position and free motion of the object being handled. Figure 11b demonstrates the procedure for collecting data from the sample object. The orientation in the XY plane (rotation) and the Z-direction were randomized. Each image data contained 256 vectors generated from 16 × 16 pixels of the sensor. The data were used as the input for training and testing the performance of the tactile recognition rate using DCNN. As part of the classification process for the model training, datasets were generated from images obtained previously from handling 20 objects (or 20 classes) by the humanoid robot hand. Those objects consisted of a battery, a remote controller, plastic tongs, a screwdriver, a coffee cup, scissors, a fixed wrench, an Allen key, a golf ball, a measuring tape, a computer mouse, a brush, an amp meter, a cola bottle, a pen, a charger, a soda bottle, a variable wrench, a water bottle, and a cream bottle. The dimensions of these objects were listed in Table 3, and the objects are shown in Figure 12. The object set was chosen similarly to [59].

### 3.4. Tactile Image Acquisition

The output voltage of each element of the tactile sensor array was read by the microcontroller board (Arduino version mega 2560). The reading was performed for each row per time by assigning the logic “1” for the row being read. On the other hand, the logic “0” was set for other rows. $\left(O_{i}=1\right),\;\left(O_{1},\;\ldots ,\;O_{i-1},\;O_{i+1},\;\ldots ,\;O_{n}=0\right)$. Then, the analog port received voltage signals (for the entire row) from the tactile sensor. The logic output was reassigned by shifting “1” to the next row, while other rows were set as “1” to obtain the next scan. This process was repeated until the entire array was scanned. Moreover, logic “0” in the multiplex circuit is grounded and sets the threshold values to cut off the noise for avoiding the crosstalk effect [35,60]. The data were sent to a computer for data processing, such as obtaining tactile images, training, and testing for image recognition. The operating system is illustrated in Figure 13a, and the image acquisition GUI is illustrated in Figure 13b.

#### 3.4.1. Tactile Image

The model testing was performed 200 times for each class, resulting in 4000 images in the database. As the DCNN employed for the experiments was the translation-invariant type, the location and orientation of each object were randomized to obtain different images of the object. Figure 14 shows the tactile images of three classes at different positions (five images per class).

#### 3.4.2. Resolution Enhancement

This study applied the resolution enhancement to resize the tactile images to achieve a suitable input resolution for DCNNs. The tactile image resolution enhancement (resize tactile image) was performed using bicubic interpolation, as shown in Equations (11)–(14), introduced by [61,62], a function to approximate the continuity of the pressure distribution over the surface by using second derivatives. As shown in Figure 15, the bicubic interpolation was calculated at 16 points surrounding the location being considered. The primary function W(*x*) of bicubic interpolation was
(11)$$W\left(x\right)=\left\{\begin{array}{c} \left(a+2\right){\left|x\right|}^{3}-\left(a+3\right){\left|x\right|}^{2}+1\\ a{\left|x\right|}^{3}-5a{\left|x\right|}^{2}+8a\left|x\right|-4a\\ 0 \end{array}\right.\begin{array}{cc} & \begin{array}{c} for\left|x\right|\leq 1\\ for1<\left|x\right|<2\\ otherwise \end{array} \end{array}$$
where the distances along the *y*-axis and *x*-axis for the interpolation B(*x,y*) (16 pixels) are designated as *K_im_* and *K_jn_*, respectively, as shown in Equation (12).
(12)$$K_{i0}=1+\mu ;K_{i1}=\mu ;K_{i2}=1-\mu ;K_{i3}=2-\mu ;$$
(13)$$K_{j0}=1+\nu ;K_{j1}=\nu ;K_{j2}=1-\nu ;K_{j3}=2-\nu ;$$

The weight coefficient is
(14)$$a_{mn}=W\left(K_{im}\right)W\left(K_{jn}\right)$$

Therefore, the interpolated image can be described by Equation (15).
(15)$$B\left(x,y\right)=\overset{3}{\underset{i=0}{\sum }}\overset{3}{\underset{j=0}{\sum }}a_{ij}A\left(x_{i},y_{i}\right)$$
where *A* is a value of the tactile image of the individual pixel of the tactile sensor, *B* is the interpolated image point, *a_ij_* represents the weight parameter, and *I*,*j* represents the *x*-axis and *y*-axis coordinates, respectively.

In Figure 14, the readings were obtained from the tactile sensor array with a 16 × 16 pixels image. The position where the object contacted the sensor appeared as a tactile image corresponding to the pressure profile. Figure 16 shows that the image quality improved significantly after applying the bicubic interpolation to enhance the image resolution. With the low-resolution image, it was challenging to indicate the object’s geometric profile. As the resolution improved, the geometric details of tactile images became more visible. These enhanced images were then used as input data for developing the recognition system.

### 3.5. DCNN

The recognition procedure applied in this study was based on the pressure profiles generated by the tactile sensor upon contacting the object. Due to image resolution, previous research proposed tactile image recognition (using the high-resolution tactile sensor) and DCNN [57,58,59]. However, using a low-resolution image for a DCNN conceivably leads to poor results [63]. Our approach used a low-resolution tactile sensor with a DCNN by using data image resolution enhancement. The following four steps were used for improving the system: (1) resolution enhancement; (2) multiple handlings for accurate prediction; (3) testing with a variety of networks to determine the appropriate network; (4) applying multimodal learning to increase the effectiveness. All steps were performed for the application of a humanoid robot hand. The DCNNs used in this study included AlexNet [64], VGG16 [65], VGG19 [65], GoogLeNet [66], ResNet18 [67], ResNet50 [67], ResNet101 [67], Place365GoogLeNet [68], InceptionV3 [69], EfficienNetB0 [70], SqueezeNet [71], InceptionResNetV2 [72], DenseNet201 [73], DarkNet19 [74], DarkNet53 [75], XceptionNet [76], NASNetMobile [77], ShuffleNet [78], and MobileNetV2 [79]. These are neural networks widely used in computer vision for object classification that has been applied for tactile object recognition by using the transfer learning method [80]. The dataset used in the DCNN training was obtained from the tactile image collection as described in Section 4.1. Tactile image signals can be represented by Equation (16) as 2D convolutions.
(16)$$\left(K*I\right)\left(i,j\right)=\underset{m,n}{\sum }K\left(m,n\right)I\left(i+n,j+m\right)$$
where *I* is a tactile image, and *K* is the kernel convolution function.

#### 3.5.1. Transfer Learning Method

The elements involving DCNNs are presented in Figure 17. This was applied with tactile object recognition by using transfer learning from ImageNet DCNN. A 20-way softmax function followed the last layer (fully connected layer) to calculate the probability distribution for 20 classes. The probability value was used for the prediction of the object, as shown in Figure 17.

#### 3.5.2. DCNN Parameters

In the tests of all datasets with DCNN, the associated training parameters were kept the same, summarized in Table 4. For each model, the initial learning rate in a range of 0.00001–0.1 was optimized based on the validation accuracy. Table 5 shows the optimized learning rate for each model. Each model was tested repeatedly ten times with the DCNN in the same dataset that contains 4000 images. The dataset was randomized and divided into a training set (50%) and a testing set (50%) in each test. The recognition rate of each model was evaluated, then, a suitable model was selected for further testing, including resolution enhancement, multiple handling, and multimodal learning.

## 4. Result and Discussion

In this study, the experiment consisted of four parts: (1) test of the recognition rate comparing tactile images obtained from object handling of the original resolution and enhanced resolution; (2) comparison of the recognition rate obtained from nineteen DCNNs; (3) the recognition rate obtained from the multimodal learning method; (4) object exploration (the effect of multiple handling of an object on the recognition rate).

### 4.1. Recognition Rate from Resolution Enhanced

In this test, AlexNet was used with an initial learning rate of 0.001. The test was performed to demonstrate the effect of image resolution on the mean recognition rate, as shown in Figure 18. The mean recognition rate was calculated from results obtained from different classes of objects. The object recognition performance increased when increasing the resolution of the tactile images. The recognition rate of approximately 77.66% was obtained with the resolution of 16 × 16. When the test was performed with resolution enhancement using bicubic interpolation, the resolution became 32 × 32 pixels and the recognition rate increased to 80.79%. Further expansion of the resolution increased the recognition rate, and the improvement became marginal when the enhanced resolution was 128 × 128 pixels. At the resolution of 512 × 512 pixels, the highest recognition rate was achieved at 84.42%. These results suggested that the efficiency of object recognition using the DCNN for object handling with the low-resolution tactile sensor can be increased by the resolution enhancement via the bicubic interpolation method. Hence, the resolution of 512 × 512 uses for further testing with other DCNN networks.

The obtained recognition results can be presented as confusion matrices. Figure 19 shows the comparison of two confusion matrices with different resolutions. When the resolution image was enhanced, prediction accuracy improved, such as predicting classes 1, 2, and 20. The prediction of classes 9, 10, and 11 was also slightly improved.

### 4.2. Recognition Results from DCNNs

In this experiment, the images passing the resolution enhancement process with a 512 × 512 pixels size were tested with nineteen DCNNs to determine the appropriate DCNN model for maximum efficiency. This DCNN model could then be applied to robotic hands and AI. Table 6 and Figure 20 compare the recognition rates obtained from different DCNNs for each tested model with the tactile image dataset. The DCNN providing the highest performance was InceptionResNetV2, with a recognition rate of 91.86%. Therefore, InceptionResNetV2 was the appropriate DCNN model for applying a tactile recognition system for humanoid robots in this work. Although SqueezeNet, EffcienNetB0, and DarkNet19 are commonly used for ImageNet, these model networks provided a relatively low recognition rate in our application, suggesting that they are not efficient for tactile images.

Figure 20 compares recognition rate, training time, and network size obtained from nineteen DCNNs. All networks were trained using MATLAB R2021a platform with Deep Learning (DL) Toolbox. In this test, a GPU (Nvidia GTX1707, US) was employed. It can be seen that AlexNet required the shortest training time (5 s/image) due to its simple architecture. On the other hand, DensNet201 required the longest training time of 108 s/image with complicated architecture. When comparing the performance of InceptionResNetV2, XceptionNet, and InceptionNetV3, InceptionResNetV2 provided the shortest training time (11.23 s/image) and the highest recognition rate. This DCNN model was then used for further testing with multiple handling of the object. Note that the network size of InceptionResNetV2 was slightly larger than that of InceptionV3 and XceptionNet.

The first three DCNNs that provided impressive performance (InceptionResNetV2, XceptionNet, and InceptionNetV3) were compared in the form of confusion matrices, as shown in Figure 21. All three models provided accurate predictions for most cases except for classes 5, 15, 16, and 20. This class was owing to the similarity of object geometry. Classes 5 (coffee cup) and 20 (cream bottle) were both cylindrical, while classes 15 (pen) and 16 (charger) were often mistaken due to the sticklike feature. For these classes, InceptionNetV2 performed slightly better than XceptionNet and InceptionNetV3.

A classification vector of InceptionResNetV2 is shown in Figure 22. It is conceivable that the recognition rate depended on the type of object. Many objects with a certain degree of similarity in shape and size can be complicated to predict. The objects with the highest recognition rate, exceeding 98.00%, included a computer mouse, a golf ball, and a fixed wrench. On the contrary, the object class associated with the recognition rate below 85.00% had a coffee cup and a cream bottle.

### 4.3. Recognition Rate from Multimodal DCNNs

In this experiment, multimodal learning [81,82] was applied to increase the effectiveness of the recognition rate. Figure 23 shows the overall experimental procedure to improve the accuracy of prediction by multimodal learning. Different pairs of models were created with six DCNNs, including InceptionResNetV2, XceptionNet, InceptionNetV3, GoogLeNet, NASNetMobile, and DensNet201, based on the performance of the recognition rate as previously presented in Section 4.2. For each pair, the InceptionResNetV2 was used as a primary model. In sequence, there were five pairs of models (InceptionResNetV2/XceptionNet, InceptionResNetV2/InceptionNetV3, InceptionRes-NetV2/GoogLeNet, InceptionResNetV2/NASNetMobile, and InceptionResNetV2/DensNet201). The recognition rate for each pair was compared with the individual network model, as shown in Table 7 and Figure 24.

Multimodal learning provided superior performance compared to the individual DCNN model. The combination of InceptionResNetV2 and XceptionNet performed exceptionally well with the highest effectiveness of 92.73% among the different bimodal pairs. This pair was 0.87% greater than that of InceptionResNetV2. The performance of other bimodal couples showed a marginal increase compared to that of InceptionResNetV2.

### 4.4. Recognition Rate from Object Exploration

This study proposes using multiple handling (object exploration) to improve the object recognition rate of the DCNN model to apply tactile object recognition as human-like. The DCNN used in this experiment was InceptionResNetV2 for the Single model and InceptionRes-NetV2/XceptionNet for Multimodal. After the system had completed learning from the training set, the system was tested with another set of images called “test set.” The output was obtained in the form of probability (*P*) for each class of object. The final output of DCNN is known as the softmax function or the normalized exponential function. This function normalizes a vector of real number logit to obtain a normalized probability, as shown in Equation (17).
(17)$$P\left(z\right)i=\frac{e^{z_{i}}}{\sum_{j=1}^{K}e^{z_{j}}}$$
where *P* is a softmax, *z* is an input vector, $e^{z_{i}}$ is a standard exponential function for input vector, *K* is a number of classes in the multiclass classifier, and $e^{z_{j}}$ is a standard exponential function for output vector.

Two methods were applied for object identification. The first method was based on the “maximum probability”. When the robot handled an object more than once, the output vector of probability was generated. Equation (18) was used as a decision model.
(18)$$P=\max\left(P\left(I_{1}\right),P\left(I_{2}\right),P\left(I_{3}\right),\ldots ,P\left(I_{n}\right)\right)$$
where *P(I_n_)* is the probability of each image.

On the contrary, the second method was based on the summation of probability. Equation (19) was used to identify the object.
(19)$$P=P\left(I_{1}\right)+P\left(I_{2}\right)+P\left(I_{3}\right)+\ldots +P\left(I_{n}\right)$$

Figure 25 shows the overall experimental procedure to improve the accuracy of prediction by multiple handling.

Results of the recognition rate are shown in Figure 26. The method of summation of probability yielded a superior performance on the recognition rate. The recognition rate of inceptionResNetV2 markedly improved from 91.86% to 96.93% when handling increased to two times. Further increasing the number of handlings increased the recognition rate at a declining rate. The recognition rate approached a plateau at 99.63% for handling greater than six times. Therefore, using the summation of probability as a criterion for object exploration for tactile image recognition using DCNN.

Figure 26 was compared in terms of confusion matrices, as shown in Figure 27. It can be seen that the error was significantly reduced by increased multiple handling. For classes 5 and 8, the method of maximum probability resulted in a slightly inaccurate prediction.

### 4.5. Discussion

The humanoid robot hand system was designed with a tactile sensor array 16 × 16 pixels, each pixel of 3.5 mm × 3.5 mm. Therefore, an active sensing area was 56.0 mm × 56.0 mm. It provided a 16  ×  16 pixels tactile image that can be applied for a tactile object recognition system when handling an object. When testing with 20 object classes using AlexNet-DCNN as the classifier, it provided a moderate recognition rate of 76.65%. Using bicubic interpolation for resolution enhancement increased the tactile image size twofold, yielding 32  ×  32 pixels tactile images. This provided an increase of 4.14% in the recognition rate. The resolution enhancement was then performed to resize 32-times. The 512  ×  512 pixels tactile images were obtained, and the recognition rate increased by 7.77%. Thus, resolution enhancement significantly improved the accuracy of the recognition rate in tactile object recognition. However, the improvement in recognition rate became marginal when the size of the enhanced image exceeded 16×–32×. Therefore, the improved tactile image size of 32× resolution is the sufficient resolution used for DCNN and is appropriate for developing a more advanced DCNN model.

The DCNN providing the best performance was InceptionResNetV2, with a recognition rate of 91.86% and training time of 11.23 s/image, followed by XceptionNet and InceptionNetV3, which provided recognition rates of 91.40% and 91.06%, respectively. The results obtained from these networks were better than AlexNet [57,58] and ResNet [59].

To further improve the recognition efficiency of a tactile object recognition system, multimodal learning was applied. The highest performance of object recognition was achieved when this method was implemented with the InceptionResNetV2 and XceptionNet. The recognition rate increased by 0.87% compared to using InceptionResNetV2 only. Based on our experimental results, the tactile sensor array developed in this work can be effective for future humanoid robot hand applications.

Another method for increasing the recognition efficiency of a tactile image recognition system by object exploration was performed by the multiple handling of objects. When the tactile image system using InceptionResNetV2 was exposed to more input information, the learning rate improved significantly, and the object prediction was more accurate. For instance, when the number of object handling increased to two times, the recognition rate increased by 4.91% for the maximum probability method and by 5.07% for the method of summation of probability. Therefore, the method of summary of possibility is appropriate for tactile object recognition for a humanoid robot, and the results obtained from DCNN are better than those for BoW [47,48,52].

Moreover, object recognition using the DCNN method can be improved by either resolution enhancement, multimodal learning, or object exploration. When these three methods are applied together, the recognition rate can be markedly increased.

## 5. Conclusions

This study used a humanoid robot palm equipped with a tactile sensor array with a 16 × 16 pixels (56.0 mm × 56.0 mm physical size) resolution to measure the pressure profile upon touching the object. Tactile images of objects were enhanced via bicubic interpolation and were used as input for the recognition system. The performance of the recognition system using different DCNN was compared in nineteen models. InceptionResNetV2 provided the ultimate 91.82% recognition rate and the maximum efficiency base on recognition rate and training time. The combined pair of InceptionResNetV2 with XceptionNet delivered the best performance with 92.73% in a tactile object recognition rate, which was better than the single InceptionResNetV2. The InceptionResNetV2 model was applied with object exploration to improve the recognition rate. The summation of probability for object exploration yielded superior performance compared to that of the maximum probability. Finally, the multimodal DCNN was employed as the learning model to improve the recognition rate of our tactile object recognition system. Therefore, the palm-size tactile sensor array developed could be effectively used to apply to a humanoid robot hand when using InceptionResNetV2 or the combination of InceptionResNetV2 and XceptionNet multimodal learning methods.

## Figures and Tables

**Figure 1 sensors-21-06024-f001:**
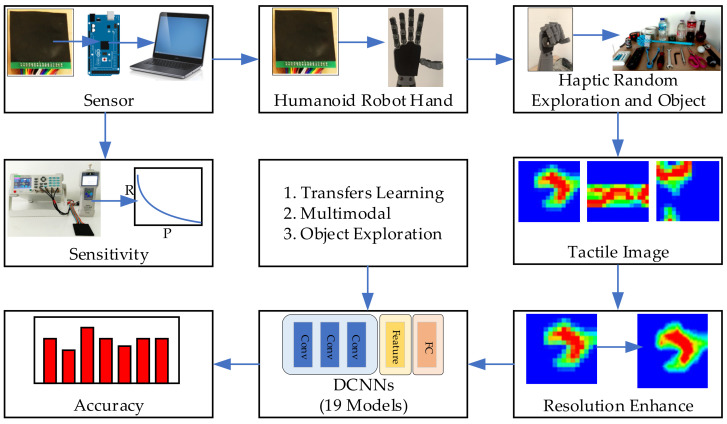
Proposed block diagram of tactile object recognition for the humanoid robot using DCNN.

**Figure 2 sensors-21-06024-f002:**
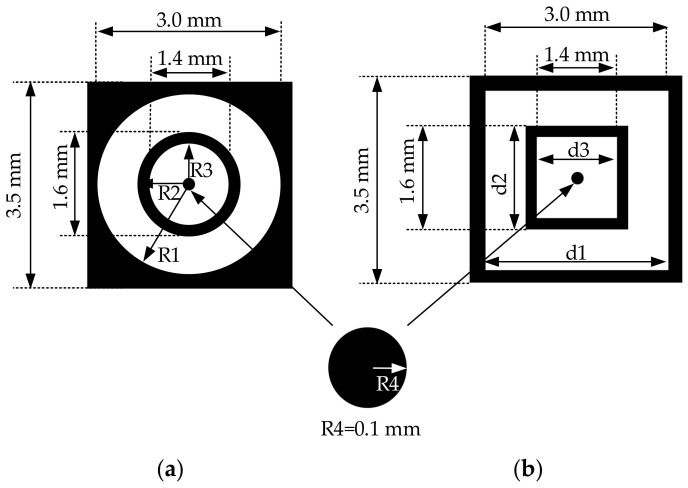
Comparison of the new circle electrode of tactile sensor and the square electrode: (**a**) A circle electrode; (**b**) A square electrode.

**Figure 3 sensors-21-06024-f003:**
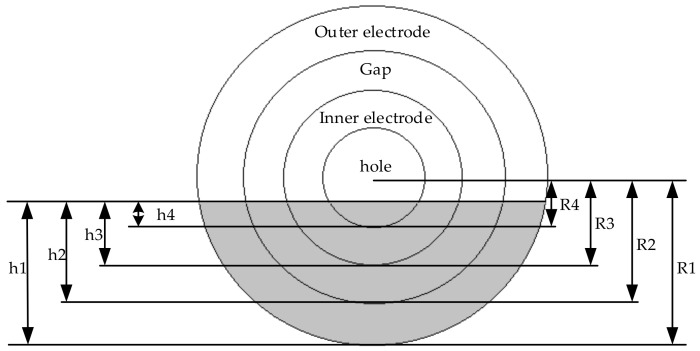
The partial contact area of the circle electrode sensor.

**Figure 4 sensors-21-06024-f004:**
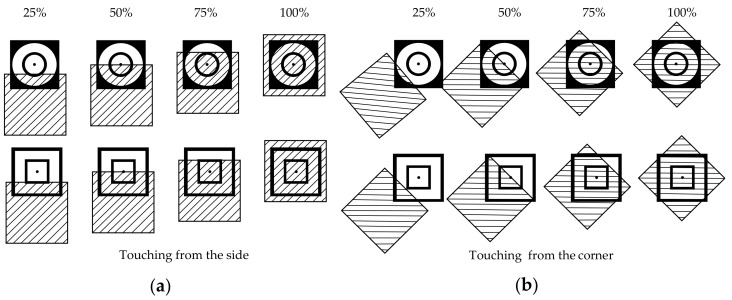
The active area of the circle and square electrode tactile sensors: (**a**) when touching the object at 25%, 50%, 75%, and 100% from the side; (**b**) when touching the object at 25%, 50%, 75%, and 100% from the corner.

**Figure 5 sensors-21-06024-f005:**
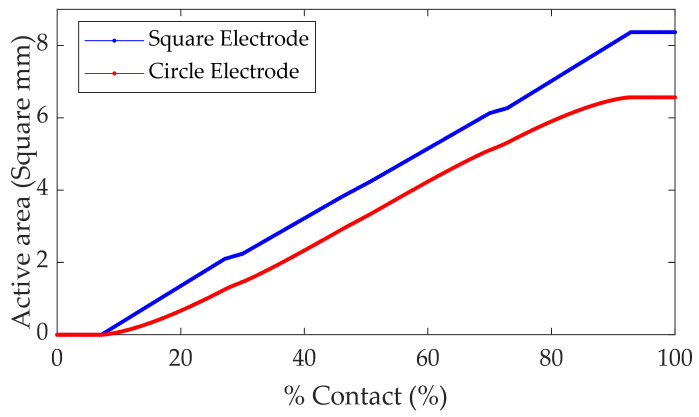
Comparison of the active area per % contact of the sensor.

**Figure 6 sensors-21-06024-f006:**
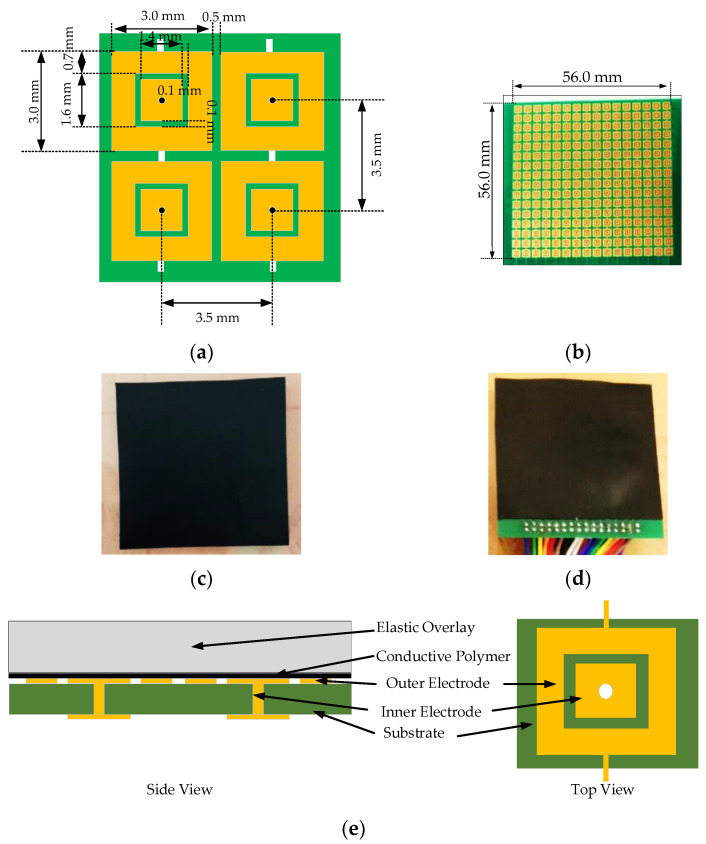
The sensor used in this research: (**a**) the size of sensor electrode; (**b**) the designed electrode and size of sensor array; (**c**) the conductive polymer sheet; (**d**) the actual sensor assembly; (**e**) layers of the sensor.

**Figure 7 sensors-21-06024-f007:**
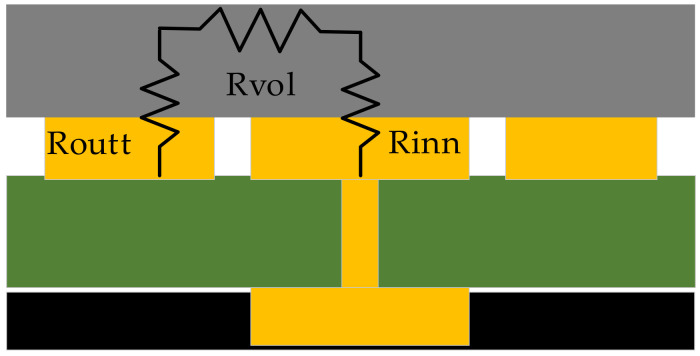
The equivalent resistance of a tactile sensor.

**Figure 8 sensors-21-06024-f008:**
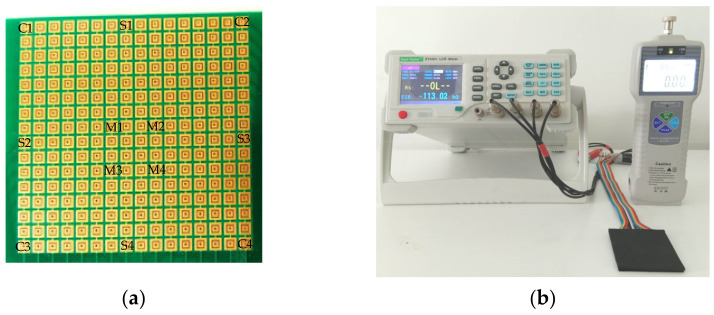
The sensor sensitivity measurement; (**a**) the sensor array with positions designated for testing; (**b**) the apparatus used for testing.

**Figure 9 sensors-21-06024-f009:**
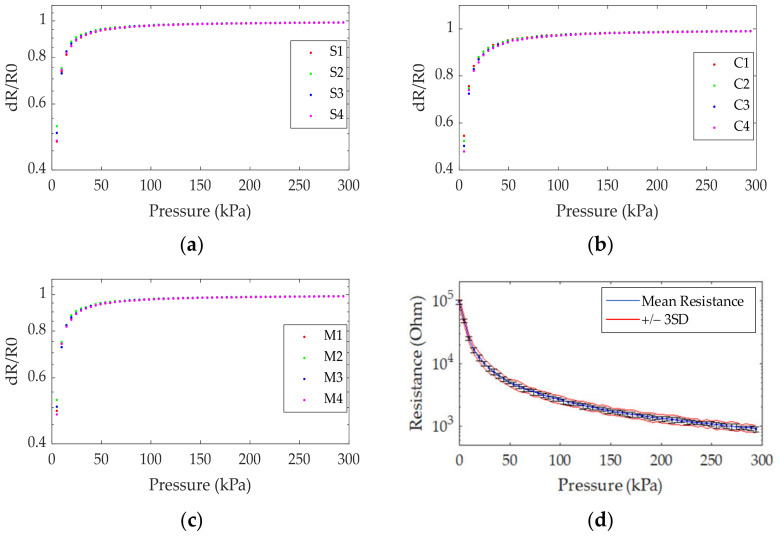
The resistance at different positions due to the pressure exerted: (**a**) sensor elements in the corners (C1, C2, C3, and C4); (**b**) sensor elements on the side (S1, S2, S3, and S4); (**c**) sensor elements in the middle (M1, M2, M3, and M4); (**d**) the average resistance of all groups.

**Figure 10 sensors-21-06024-f010:**
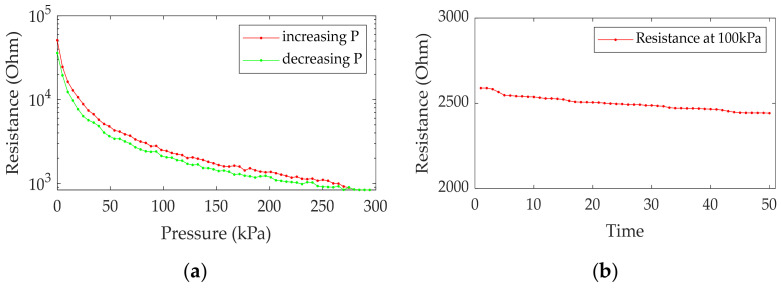
The sensor properties: (**a**) Sensor hysteresis; (**b**) Sensor reproducibility.

**Figure 11 sensors-21-06024-f011:**
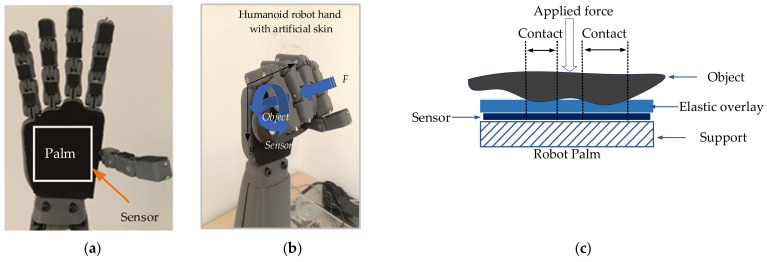
Experimental setup: (**a**) A humanoid robot hand equipped with a tactile sensor array on the palm; (**b**) A humanoid robot hand during object handling; (**c**) Force-sensing transducer of the tactile sensor array.

**Figure 12 sensors-21-06024-f012:**
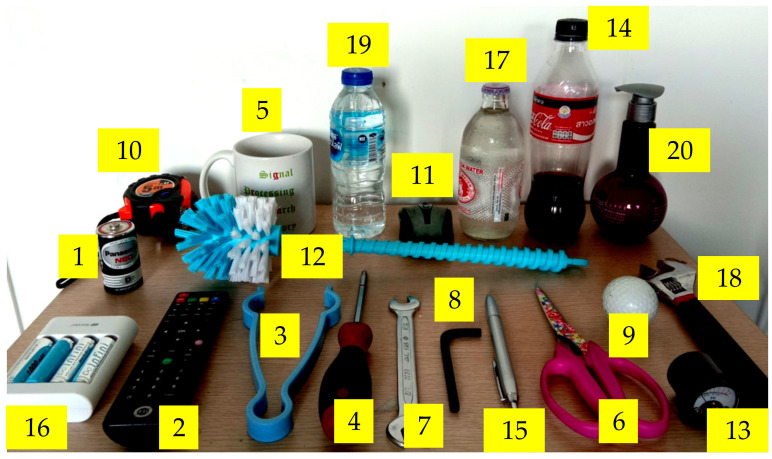
Objects used for model training. (1) battery, (2) remote controller, (3) plastic tongs, (4) screwdriver, (5) coffee cup, (6) scissors, (7) fixed wrench, (8) Alan key, (9) golf ball, (10) measuring tape, (11) computer mouse, (12) brush, (13) amp meter, (14) cola bottle, (15) pen, (16) charger, (17) soda bottle, (18) variable wrench, (19) water bottle, (20) cream bottle.

**Figure 13 sensors-21-06024-f013:**
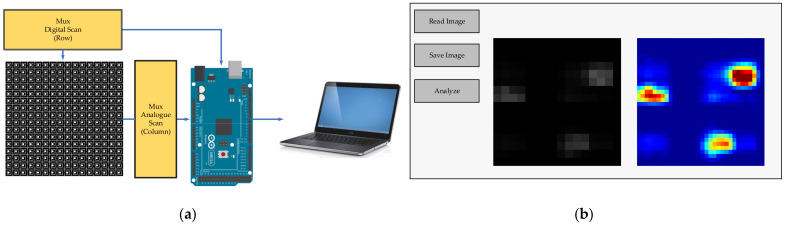
Tactile image acquisition system: (**a**) The schematic diagram of the tactile sensor array for image acquisition; (**b**) Image acquisition GUI.

**Figure 14 sensors-21-06024-f014:**
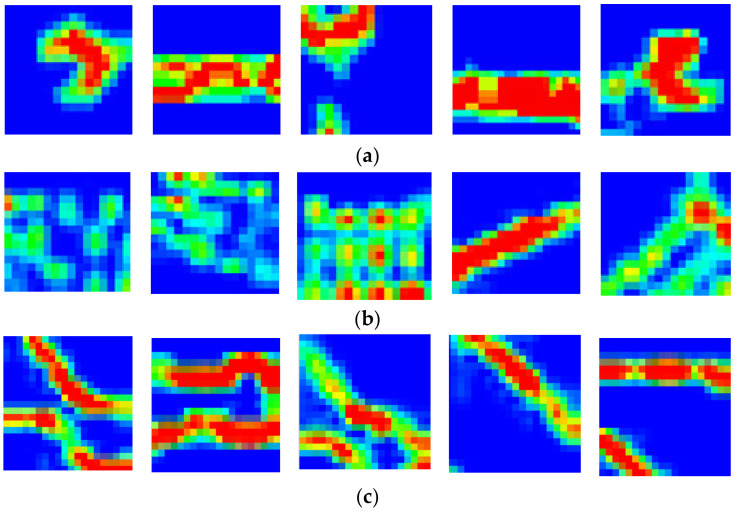
Sample tactile images: (**a**) a fixed wrench; (**b**) a remote controller; (**c**) plastic tongs.

**Figure 15 sensors-21-06024-f015:**
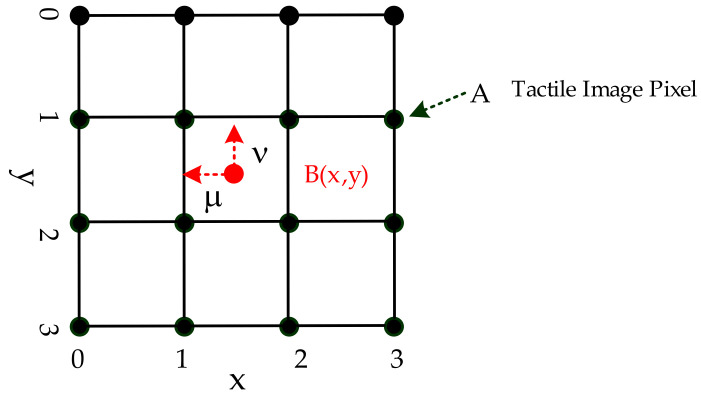
Bicubic interpolation grids.

**Figure 16 sensors-21-06024-f016:**
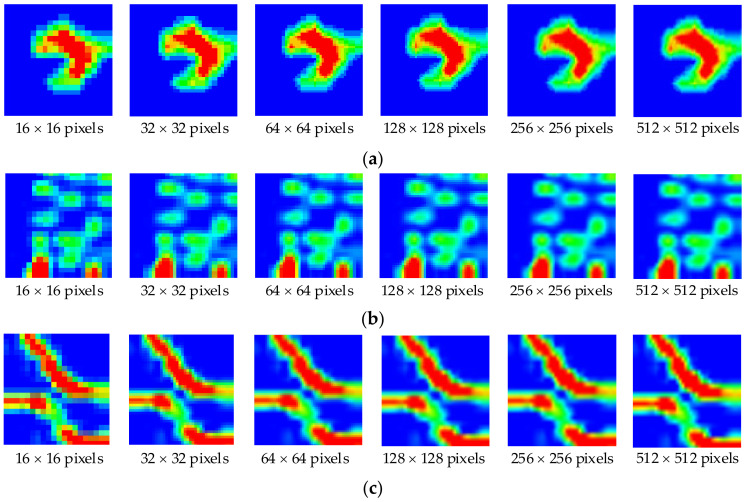
Enhanced tactile images for different objects: (**a**) fixed wrench; (**b**) remote controller; (**c**) plastic tongs.

**Figure 17 sensors-21-06024-f017:**
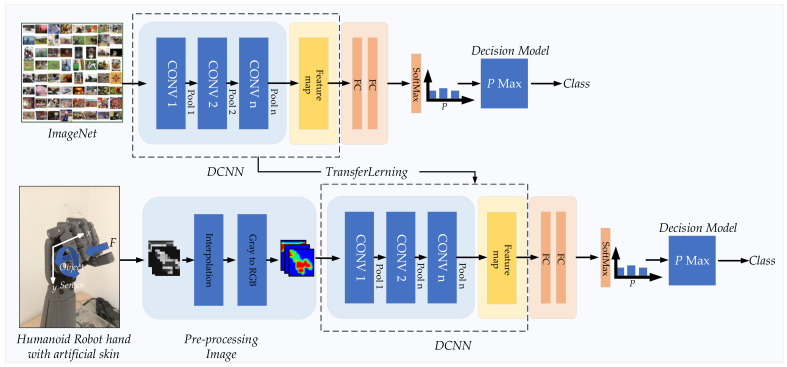
DCNN Transfer Learning method was used in this study.

**Figure 18 sensors-21-06024-f018:**
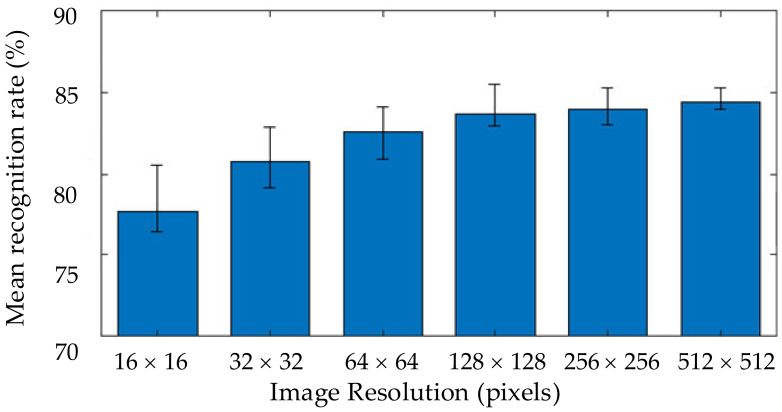
The recognition rate for images obtained with different resolutions.

**Figure 19 sensors-21-06024-f019:**
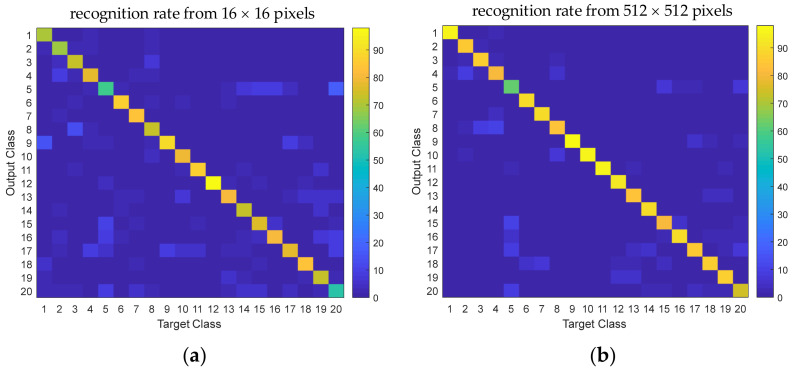
Confusion matrices were obtained using DCNN for 20 classes of objects: (**a**) Tactile image resolution 16 × 16 pixels without resolution enhancement; (**b**) Tactile image with enhanced resolution of 512 × 512 pixels.

**Figure 20 sensors-21-06024-f020:**
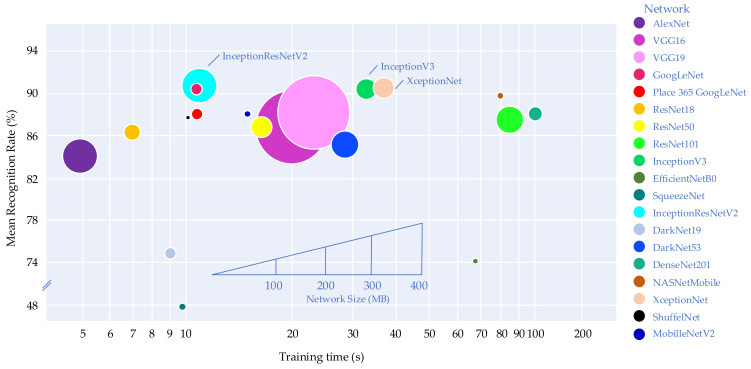
The recognition rate of each DCNN from 512 × 512 pixels resolution.

**Figure 21 sensors-21-06024-f021:**
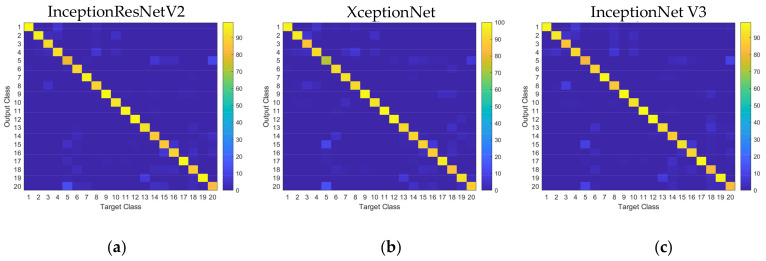
Confusion matrices obtained from (**a**) InceptionResNetV2 (**b**) XceptionNet (**c**) InceptionNetV3.

**Figure 22 sensors-21-06024-f022:**
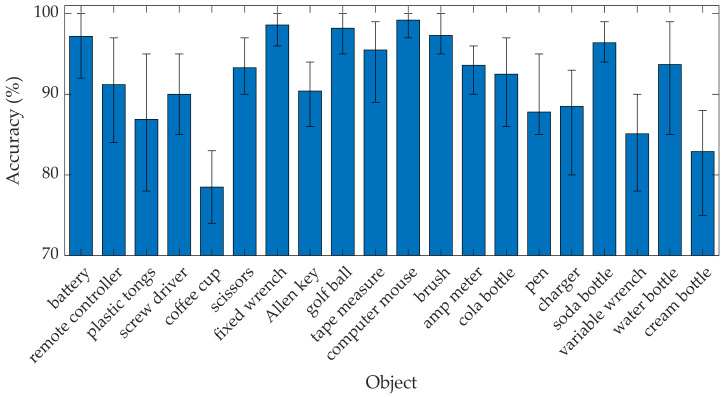
Recognition rate of InceptionResNetV2.

**Figure 23 sensors-21-06024-f023:**
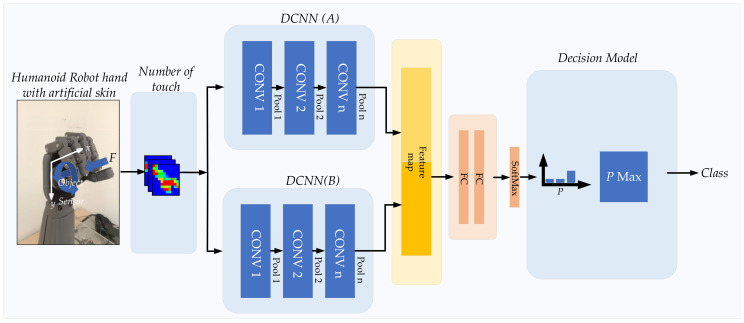
Object recognition using multimodal DCNNs.

**Figure 24 sensors-21-06024-f024:**
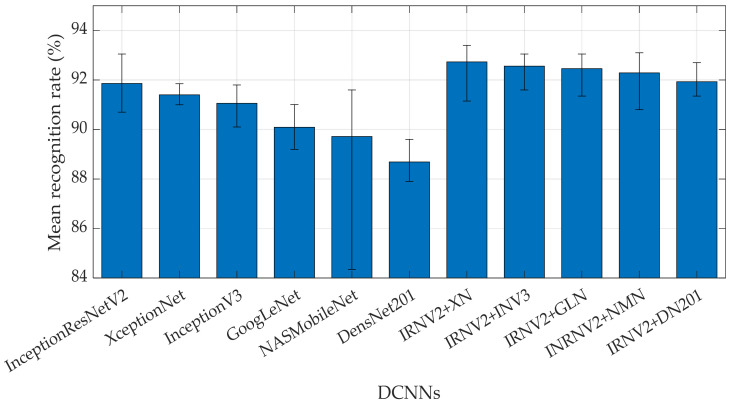
The recognition rate of single DCNNs and Multimodal DCNNs.

**Figure 25 sensors-21-06024-f025:**
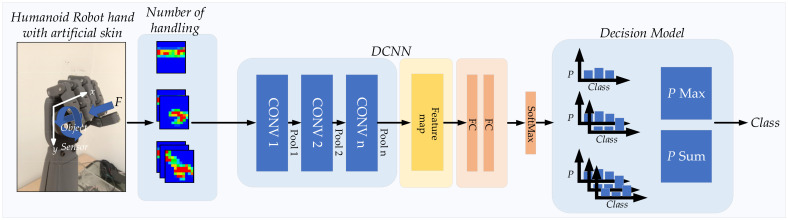
Object recognition from object exploration.

**Figure 26 sensors-21-06024-f026:**
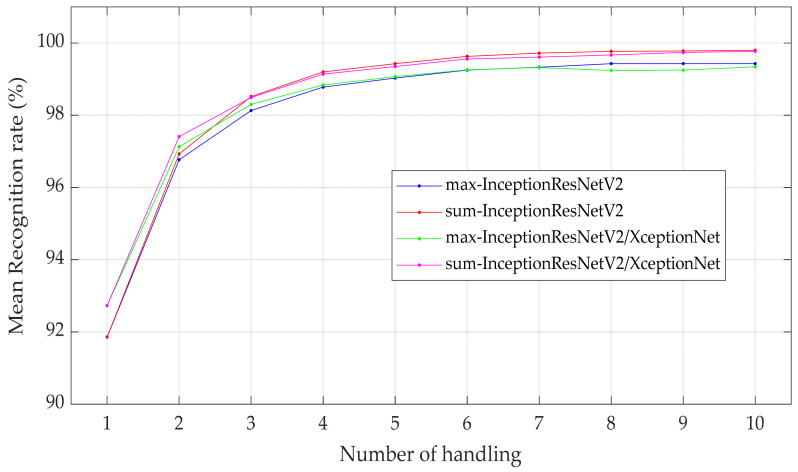
The recognition rate for multiple handling using a maximum probability method and summation of probability method.

**Figure 27 sensors-21-06024-f027:**
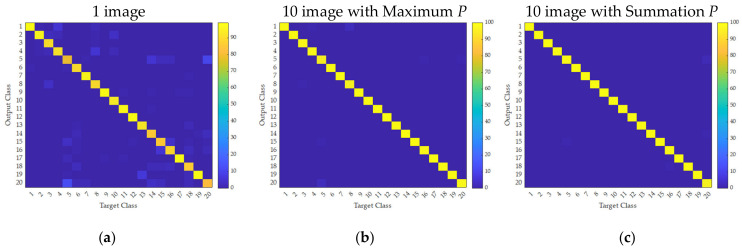
Confusion matrices obtained from multiple handling of the object using a DCNN: (**a**) once; (**b**) 10 times based on the maximum of probability; (**c**) 10 times based on the summation of probability.

**Table 1 sensors-21-06024-t001:** Piezoresistive tactile sensor array for the anthropomorphic robot.

Year	Sensor Resolution (Pixel)	Physical Size (mm)	Sensor Size (mm)	Transducer	Application
2007 [32]	9 × 8	20 × 30	1.8 × 3.4	Conductive rubber	Finger
2009 [33]	25 × 25	150 × 150	5.0 × 5.0	Organic resistance	Flex e-Skin
2009 [34]	5 × 5	32 × 32	50 mm^2^	FSR	Touch sensing
2010 [35]	16 × 16	50 × 50	3.44 mm^2^	Conductive polymer	Flex e-Skin
2011 [36]	8 × 8	20 × 20	2.5 × 2.5	Conductive rubber	Gripper Finger
2011 [37]	8 × 16	70 × 120	3.0 × 3.0	Nickel power + PDMS	Flex e-Skin
2104 [38]	8 × 8	-	3.0 × 3.0	P(VDF-TrFE), MWCNT/PDMS	Flex e-Skin
2014 [39]	10 × 10	20 × 20	1.3 × 1.3	Nanoparticles of carbon and silica	Finger
2015 [40]	3 × 3	50 × 50	10 × 10	Conductive pillars	Flex e-Skin
2017 [41]	8 × 8	300 × 220	37.5 × 2.5	Conductive polymer	e-Skin
2017 [42]	8 × 8	100 × 100	480 mm^2^	Polymer composite	e-Skin
2018 [43]	1 × 1	20 × 25	20 × 25	Conductive polymer	Finger tip
2019 [44]	4 × 4	10 × 10	1900 mm^2^	Polymer composite	e-Skin

**Table 2 sensors-21-06024-t002:** Tactile image recognition for robot application.

Year	Sensor Resolution (Pixel)	Number of Class	Descriptor	Classification Method	Number of Grasps	Recognition Rate (%)
2009 [47]	6 × 14 (2ea)	21	All Data Vector	BoW	10	84.6
2011 [36]	8 × 8 (2ea)	10	Mean, SD	kNN	1	92.0
2011 [48]	6 × 6	5	Polar Furrier	PCA, BoW	>50	90.0
2012 [49]	24 × 16	4	Maximum Vector	PCA, kNN	1	81.4
2012 [50]	5 × 9, 12 × 10	4	3 × 3 Segmentation	ANN	1	91.0
2013 [51]	32 × 32	10	Haar Wavelet	kNN	1	86.0
2014 [39]	10 × 10	12	K-PCA, FD, GE	MKL-SVM	1	85.54
2015 [52]	6 × 14	18	Segmentation SIFT	BoW	15	89.9
2015 [53]	16 × 16	25	Reduced Vector	ANN	1	96.0
2015 [54]	108+133	20	All Data Vector	DNN	1	91.1
2016 [55]	8 × 3 (3ea), 7 × 4	7	All Data Vector	KSC	1	94.0
2016 [56]	6 × 14	20	Zernike moment	iCLAP	20	85.36
2017 [57]	768	2	AlexNet-DCNN	DCNN	1	98.3
2017 [58]	28 × 50	8	AlexNet-DCNN	DCNN	1	91.6
2019 [59]	28 × 50	22	ResNet-DCNN	DCNN	1	95.36

**Table 3 sensors-21-06024-t003:** Dimensions of objects used in the model training.

Class	Object	Size
1.	Battery	∅ = 33 mm, H = 60 mm
2.	Remote controller	L = 172 mm, W = 45 mm, H = 15 mm
3.	Plastic tongs	L = 165 mm, W = 85 mm, H = 15 mm
4.	Screwdriver	∅ = 38 mm, L = 220 mm
5.	Coffee Cup	∅ = 85 mm, W = 120 mm, H = 95 mm
6.	Scissors	L = 18 mm, W = 80 mm, H = 12 mm
7.	Fixed wrench	L = 168 mm, W = 30 mm, H = 6 mm
8.	Allen key	L = 90 mm, W = 35 mm, H = 6 mm
9.	Golf ball	∅ = 40 mm
10.	Measuring tape	∅ = 70 mm, W = 92 mm, H = 47 mm
11.	Computer mouse	L = 100 mm, W = 50 mm, H = 35 mm
12.	Brush	∅ = 80 mm, L = 390 mm
13.	Amp meter	∅ = 35 mm, L = 35 mm
14.	Cola bottle	∅ = 60 mm, H = 22 mm
15.	Pen	∅ = 12 mm, L = 147 mm
16.	Charger	L = 90 mm, W = 70 mm, H = 22 mm
17.	Soda Bottle	∅ = 65 mm, H = 155 mm
18.	Variable wrench	L = 200 mm, W = 60 mm, H = 14 mm
19.	Water bottle	∅ = 60 mm, H = 175 mm
20.	Cream bottle	∅ = 75 mm, H = 160 mm

**Table 4 sensors-21-06024-t004:** Training parameters.

Parameters	Values
NumEpochs	30
NumBatchSize	16
Momentum	0.9
LearnRateDropFactor	0.1
LearnRateDropPeriod	8

**Table 5 sensors-21-06024-t005:** The initial learning rate of each DCNN.

Initial Learning Rate	0.00001	0.0001	0.001	0.01
DCNN Model	SqueezeNet	VGGNet16 VGGNet19 DarkNet19 DarkNet53	AlexNet GoogLeNet Place365GoogleNet EfficienNetB0	ResNet18 ResNet50 ResNet101 InceptionV3 InceptionResNetV2 DensNet201 XceptionNet NASNetMobile ShuffleNet MobileNetV2

**Table 6 sensors-21-06024-t006:** The recognition rate of each DCNN from 16 × 16 pixels and 512 × 512 pixels resolution.

Model	Accuracy of 16 × 16 Pixels				Accuracy of 512 × 512 Pixels			
	Min	Max	Mean	SD	Min	Max	Mean	SD
AlexNet	76.40	80.55	77.66	1.27	84	85.2	84.42	0.40
VGG16	79.95	81.85	80.92	0.58	85.7	88.35	86.94	0.78
VGG19	80.10	84.20	81.84	1.11	86.4	88.6	87.62	0.66
GoogLeNet	85.15	86.30	85.57	0.36	89.2	91.01	90.09	0.59
ResNet18	80.05	83.00	81.36	0.99	85.25	87.25	86.5	0.61
ResNet50	81.55	84.60	82.87	0.95	86.1	88.55	87.59	0.82
ResNet101	81.75	85.40	83.61	1.12	86.75	89.65	88.07	0.85
Place365GoogLeNet	74.80	81.00	78.40	1.84	87.2	89.75	88.56	0.80
InceptionNetV3	85.25	87.45	86.47	0.75	90.1	91.8	91.06	0.50
EfficienNetB0	71.25	76.95	74.18	2.01	71.25	76.95	74.18	2.01
SqueezeNet	39.02	41.50	40.21	1.71	45.5	51.6	48.33	1.71
InceptionResNetV2	84.55	87.20	85.15	0.91	90.7	93.05	91.86	0.70
DarkNet19	61.60	68.05	65.01	1.74	75	78.8	77.36	1.17
DarkNet53	77.30	81.55	79.44	1.43	83.85	86.2	85.42	0.90
DenseNet201	82.40	86.10	83.79	1.15	87.9	89.6	88.69	0.56
XceptionNet	82.45	86.15	84.48	1.06	91	91.85	91.4	0.29
NASNetMobile	85.10	87.35	86.36	0.69	84.35	91.6	89.72	2.08
ShuffleNet	81.65	83.80	82.65	0.78	87.25	88.8	88.16	0.55
MobileNetV2	82.30	85.55	83.92	0.92	87.35	89.9	88.23	0.73

**Table 7 sensors-21-06024-t007:** The recognition rate of multimodal DCNNs.

Model	Accuracy			
	Min	Max	Mean	SD
InceptionResNetV2/XceptionNet	91.85	93.35	92.73	0.51
InceptionResNetV2/InceptionNetV3	91.60	93.05	92.56	0.49
InceptionResNetV2/GoogLeNet	91.35	93.05	92.46	0.47
InceptionResNetV2/NASNetMobile	90.80	93.10	92.29	0.82
InceptionResNetV2/DensNet201	91.35	92.70	91.93	0.43

## Data Availability

Not applicable.

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
