# Peer review of "Tactile Object Recognition for Humanoid Robots Using New Designed Piezoresistive Tactile Sensor and DCNN"

_sensors, 2021, doi:10.3390/s21186024_

Round 1
Reviewer 1 Report
The main research content of this article: a tactile sensor array was developed. Its all electrodes are at a same PCB , which simplified the wiring. The sensor sample was applied on the palm of the humanoid robots for object recognition. Based on the sensor, an algorithm for tactile image recognition was proposed, and 19 DCNN models was used for training and testing. The results of Experiments show that the object recognition rate can be improved through resolution enhancement, multi-modal learning or object exploration. When the three are applied at the same time, the recognition rate is significantly improved. There are some problems should be noted,
1. The abstract has written an assessment of sensitivity, but there is no calculation of sensitivity in the article. It is not adequate only providing the resistance varied with acting force.
2. The partial contact area of the circular electrode sensor in Figure 3 is not clearly shown, and it is not known whether it is the inner electrode or the outer electrode or the entire electrode.
3. The label in Figure 3 is θ and the formula (6) and (7) uses α.
4. How was the formula (8) deduced? Why all the resistance values in Figure 8(c) are calculated by formula (8) is not explained.
5. It is suspect that the Formula (9) of Bicubic function constructed by bicubic interpolation is incorrect.
6. The overall format of the whole article is messy.
7. It looks that the resistance of the piezoresistive tactile sensor units is infinite when there is no force acting on. Why? whether it is influential for testing?
8. The palm of the humanoid robot in the article is a invariable plane. Normally there are joints on the human palm, so the sensor array may be not very suitable for object recognition.
Reviewer 2 Report
Overall the paper has merit but the following comments are suggested for improving the paper and make more understandable.
- Some of the information in the paper could be moved to supplementary information to reduce the size of the manuscript and enable reader focus on main contents while reading it (for instance the comparison of the circle and the square electrodes in Section 3.1). The paper also contains too many figures, some of these figures could be clubbed together.
- Please correct the unit of 50,000 ohm/cm to 50,000 ohm/cm2. (Line 206, page 7) and also, 18 um to 18 µm (Line 208, page 7).
- Correct the spelling of electrode in Figure 6, font in Figure 8 could also be improved.
- Will be good to quantify the y-axis of Figure 8 as change in resistance rather than the respective resistance values. The x-axis should preferably be represented in Pascals for easy comparison with existing sensors.
- Did the authors observe any crosstalk in the array, and how did you manage this?
- How reliable and repeatable is the output data coming from the sensors? A loading and unloading cyclic measurements will be useful to understand how stable the output of the sensor is? What’s the performance of the sensor with and without bending?
